# Transcriptomic Profiles of CD47 in Breast Tumors Predict Outcome and Are Associated with Immune Activation

**DOI:** 10.3390/ijms22083836

**Published:** 2021-04-07

**Authors:** María del Mar Noblejas-López, Mariona Baliu-Piqué, Cristina Nieto-Jiménez, Francisco J. Cimas, Esther C. Morafraile, Atanasio Pandiella, Ángel L. Corbi, Balázs Győrffy, Alberto Ocaña

**Affiliations:** 1Translational Oncology Laboratory, Translational Research Unit, Albacete University Hospital, 02008 Albacete, Spain; mariadelmar.noblejas@uclm.es (M.d.M.N.-L.); franciscojose.cimas@uclm.es (F.J.C.); 2Centro Regional de Investigaciones Biomédicas, Castilla-La Mancha University (CRIB-UCLM), 02008 Albacete, Spain; 3Experimental Therapeutics Unit, Medical Oncology Department, Hospital Clínico San Carlos (HCSC), Instituto de Investigación Sanitaria (IdISSC) and CIBERONC, 28029 Madrid, Spain; mariona.baliu@salud.madrid.org (M.B.-P.); cnietoj@salud.madrid.org (C.N.-J.); esther.cabanas@salud.madrid.org (E.C.M.); 4Instituto de Biología Molecular y Celular del Cáncer (IBMCC-CIC), 37007 Salamanca, Spain; atanasio@usal.es; 5Instituto de Investigación Biomédica de Salamanca (IBSAL), 37007 Salamanca, Spain; 6CIBERONC, 37007 Salamanca, Spain; 7Consejo Superior de Investigaciones Científicas (CSIC), 37007 Salamanca, Spain; 8Centro de Investigaciones Biológicas (CIB), Departamento de Biología Celular, Consejo Superior de Investigaciones Científicas (CSIC), 28029 Madrid, Spain; acorbi@cib.csic.es; 9Department of Bioinformatics, Semmelweis University, H-1094 Budapest, Hungary; gyorffy.balazs@ttk.mta.hu; 102nd Department of Pediatrics, Semmelweis University, H-1094 Budapest, Hungary; 11TTK Lendület Cancer Biomarker Research Group, Institute of Enzymology, H-1117 Budapest, Hungary

**Keywords:** CD47, immune activation, pro-tumoral macrophages, immunotherapy, breast cancer

## Abstract

Targeting the innate immune system has attracted attention with the development of anti- CD47 antibodies. Anti-CD47 antibodies block the inhibition of the phagocytic activity of macrophages caused by the up-regulation of CD47 on tumor cells. In this study, public genomic data was used to identify genes highly expressed in breast tumors with elevated CD47 expression and analyzed the association between the presence of tumor immune infiltrates and the expression of the selected genes. We found that 142 genes positively correlated with CD47, of which 83 predicted favorable and 32 detrimental relapse-free survival (RFS). From those associated with favorable RFS, we selected the genes with immunologic biological functions and defined a CD47-immune signature composed of PTPRC, HLA-E, TGFBR2, PTGER4, ETS1, and OPTN. In the basal-like and HER2+ breast cancer subtypes, the expression of the CD47-immune signature predicted favorable outcome, correlated with the presence of tumor immune infiltrates, and with gene expression signatures of T cell activation. Moreover, CD47 up-regulated genes associated with favorable survival correlated with pro-tumoral macrophages. In summary, we described a CD47-immune gene signature composed of 6 genes associated with favorable prognosis, T cell activation, and pro-tumoral macrophages in breast cancer tumors expressing high levels of CD47.

## 1. Introduction

Administration of inhibitors of immunosuppressive signals has become an effective therapeutic strategy for the treatment of different types of cancer [1,2]. This approach has clearly modified the concept of cancer therapeutics, opening the door for the exploitation of the immune system to treat oncogenic processes [3]. In contrast to classical chemotherapy or targeted agents, immunotherapy aims to stimulate the patient’s own immune system to attack tumor cells, therefore inducing long-lasting responses [3].

Programmed cell death 1 (PD1) and its ligand, PD-L1, are negative regulators of T cell activation that act as ‘checkpoint molecules’ [4]. Targeting PD1 and PD-L1 for the treatment of cancer enhances the response of T cells against the tumor [1,2,3,5]. Unfortunately, not all treated patients respond to checkpoint inhibitors, but immune-activated tumors, including those with high PD-L1 expression, are more prepared to orchestrate an adequate immune response following checkpoint blockade [6,7]. In this context, several studies have explored the potential of genomic signatures to predict outcomes and response to checkpoint inhibitors by identifying immune-activated tumors [8,9,10].

Exploiting immunotherapy as a therapeutic tool has mainly focused on the adaptive immune system to induce and boost an efficient T cell response [3]. Targeting innate immunity has recently attracted attention as a potential therapeutic option for many types of human cancers [11,12]. Macrophages, one of the components of the innate immune system, can contribute to the elimination of tumor cells by phagocytosis and contact-dependent and independent killing. Of note, two subsets of macrophages have been described: The M1 subtype, which exhibits a pro-inflammatory phenotype and displays anti-tumoral activities and phagocytic functions, and the M2 subtype, closely related to the so-called tumor-associated macrophages (TAMs), which has potent anti-inflammatory and tissue-repair (fibrotic) functions and promotes tumor progression [13].

Inhibitory signals also control macrophages activation. The signal-regulatory protein α (SIRPα) is an inhibitory receptor that presents immunoreceptor tyrosine-based inhibition motifs (ITIMs) [14,15]. SIRPα is expressed not only on macrophages but also on dendritic cells (DC) and neutrophils [16,17]. SIRPα plays an inhibitory role when activated by its ligand CD47. This interaction generates a “do not eat me” signal that prevents phagocytosis. Cancer cells may escape the immune surveillance of macrophages by the upregulation of CD47 expression [17,18].

Strategies to block this inhibitory pathway are under evaluation and aim at mimicking the success achieved with PD1 and PD-L1 inhibitors. Inhibition of the SIRPα-CD47 axis will enable macrophages to phagocytize and eliminate tumor cells in an efficient manner [16,17]. Currently, more than twenty early-stage clinical studies evaluating antibodies against the SIRPα-CD47 axis are ongoing [17]. Following the approach with checkpoint inhibitors evaluating PD-L1 expression, some of these new studies explore the presence of CD47 in relation to clinical efficacy. However, it is expected that the mere expression of this marker, like for PD-L1 expression, would not completely identify responder tumors.

In this study, we aimed to identify genomic correlates associated with the expression of CD47 in breast cancer to get insights into the immunologic characteristics of those tumors.

## 2. Results

### 2.1. Identification of Genes Expressed in Breast Tumors with High Expression of CD47

We used public genomic data to identify genes highly associated with CD47 expression at mRNA level in breast cancer tumors (*n* = 1764). Thereafter, we analyzed the transcripts which were positively (Spearman correlation, SC > 0.4 and *p* < 0.05) and negatively (SC < −0.4 and *p* < 0.05) correlated with CD47 expression. We identified 142 genes with a positive and five genes with a negative correlation with CD47 expression (Figure 1a).

We next explored the association of the selected gene transcripts with patient outcome, in terms of relapse-free survival (RFS), in 1764 breast cancer patients from all subtypes. From the 142 genes positively correlated with CD47, 83 genes (58.5%) predicted favorable RFS, and 32 genes (22.5%) were predicted to be detrimental to RFS. Of note, no association with survival was observed for 27 genes (19%). The five genes negatively correlated with CD47 were associated with a favorable prognosis (Figure 1a). A complete list of the identified genes is shown in Appendix A.

The get insights into the biological functions of the identified genes, we used the gene set enrichment analysis tool Enrichr (http://amp.pharm.mssm.edu/Enrichr/, accessed on 20 March 2020) [19]. For those genes correlated with detrimental prognosis, no immunologic-related functions were found (Appendix A), neither for the five genes negatively correlated with CD47 expression (Appendix A). For the genes positively correlated with CD47 and associated with favorable RFS, 17 biological processes with a *p* < 0.01 value were detected (Figure 1b). As the main goal of this project was to explore immunological correlates associated with the expression of CD47, we focused only on those biological functions related to immunology. From those, we selected only those genes from biological processes related to the immune system: (i) regulation of T cell-mediated cytotoxicity (PTPRC, HLA-E, and TGFBR2) and (ii) positive regulation of defense response (PTGER4, ETS1, and OPTN) (Figure 1b). Other biological functions included ‘cell protein modification’, ‘endosome organization’, ‘protein localization to plasma membrane’, ‘response to peptide stimulus’, and ‘positive regulation of biosynthetic process’, among others (Figure 1b). A complete list of the biological functions of the proteins codified by PTPRC, HLA-E, TGFBR2, PTGER4, ETS1, and OPTN genes obtained via UniProt is shown in Appendix A.

### 2.2. CD47-Immune Signature is Associated with Favorable Prognosis in Breast Cancer, Especially for the BASAL-Like and HER2+ Subtypes

Each gene individually, PTGER4, ETS1, PTPRC, HLA-E, TGFBR2, and OPTN was associated with favorable outcomes (RFS and OS) in a statistically significant manner (Figure 1c); although, some did not predict better than the just the expression of CD47. However, given the fact that CD47 expression has been described as a negative regulator of the anti-tumoral action of macrophages [20,21], we aimed to explore the association of the CD47-immune signature composed by PTGER4, ETS1, PTPRC, HLA-E, TGFBR2, and OPTN with clinical outcomes. Using the exploratory dataset which includes more than 1764 patients with RFS data, and 626 patients with OS information, we found that the combination of PTGER4, ETS1, PTPRC, HLA-E, TGFBR2, and OPTN predicted favorable RFS (HR = 0.65; CI = 0.55–0.76; *p* = 1.2 × 10^−7^ and OS (HR = 0.53; CI = 0.39–0.73; *p* = 5.6 × 10^−5^) in breast cancer (Figure 1d). This prediction was better than single gene prediction, including CD47, and displayed a very low false-discovery rate (FDR). 

Next, we explored if clinical outcomes could differ based on different breast cancer subtypes, as immune surveillance in each tumor subtype can be substantially different. In line with this heterogeneity, the prediction capacity of each gene varied, being the basal-like and HER2+ subtypes those in which the majority of genes predicted a favorable outcome, particularly for OS (Figure 2a). A similar correlation was observed for most of the genes in the basal-like, but not for the HER2 subtype, in the Molecular Taxonomy of Breast Cancer International Consortium (METABRIC) study (Figure 2b). For the combined signature, we found that differences in outcome were more evident for the basal-like and HER2+ breast cancer subtypes for both RFS and OS (Figure 2c,d). In the basal-like subgroup, the expression of the CD47-immune signature predicted favorable outcome for RFS (HR = 0.4; CI = 0.29–0.56; *p* = 1.2 × 10^−8^) and OS (HR = 0.23; CI = 0.12–0.44; *p* = 1 × 10^−6^) (Figure 2c). In the HER2+ subgroup a similar association was identified for RFS (HR = 0.43; CI = 0.27–0.68; *p* = 0.00021) and OS (HR = 0.25; CI = 0.11–0.57; *p* = 0.00034) (Figure 2d). 

Given the exploratory nature of this cohort, we next analyzed these results using a confirmatory dataset. To do so, we used the METABRIC study that involved more than 1988 patients (PMID: 22522925). This dataset only provides information about OS. Using the validation cohort, we confirmed that for basal-like (HR = 0.54; CI = 0.4–0.73; *p* = 4.4 × 10^-5^) and HER2+ (HR = 0.6; CI = 0.38–0.95; *p* = 0.025) tumors the CD47-immune signature predicted favorable OS (Figure 2e,f). Altogether, this data demonstrates that the prediction observed was particularly strong and reproducible in the basal-like subtype. 

In luminal B tumors, the association between the CD47-immune signature and patient outcome was statistically associated with outcome (RFS: HR = 0.5; CI = 0.35–0.69; *p* = 3.2 × 10^-5^; OS: HR = 0.39; CI = 0.19–0.79; *p* = 0.0072) (Figure 2g). This result was confirmed using the validation cohort (OS: HR = 0.82; CI = 0.68–0.99; *p* = 0.038) (Figure 2i). However, for the more frequent breast cancer subtype, the luminal A subgroup, no association was observed in either of the two cohorts: for the exploratory cohort RFS: HR = 0.81; CI = 0.63–1.03; *p* = 0.09; OS: HR = 0.65; CI = 0.37–1.16; *p* = 0.14 (Figure 2h) and for the validation cohort: OS: HR = 0.86; CI = 0.7–1.05; *p* = 0.14 (Figure 2j). 

### 2.3. CD47-Immune Signature Correlated with the Presence of Immune Infiltrates in Basal-Like and HER2+ Breast Tumors

The expression of the six genes composing the CD47-immune signature was associated with low tumor purity in all breast cancer subtypes analyzed (Figure 3), suggesting a high infiltration of non-tumor cells. For the basal-like subtype a positive correlation (partial correlation (pc) > 0.4) was observed for PTGER4, ETS1, OPTN, PTRC, and HLA-E for DCs, neutrophils, and CD4+ T cells. PTPRC, HLA-E, and TGFBR2 were also highly correlated with the presence of CD8+ T cells (pc > 0.5). No association was observed with macrophages, with the exception of TGFBR2 (pc > 0.5) (Figure 3b). For the HER2+ subtype, a positive correlation (pc > 0.4) was observed between ETS1, PTPRC, and HLA-E and CD8+ T cells, CD4+ T cells, neutrophils, and DCs, with ETS1 and PTPRC showing a pc > 0.7. Expression of TGFBR2 was again linked with a high presence of macrophages (pc > 0.6), while no association with macrophages was identified for the other genes (Figure 3c). For the luminal subtype, a positive association was observed for most genes, but more significantly for PTPRC and DC, neutrophils, CD4+ T cells, and CD8+ T cells (Figure 3d). 

### 2.4. CD47-Immune Signature is Associated with Markers of T Cell Activation and Antigen Presentation

Next, we explored the association between the genes included within the CD47-immune signature and genes that encode for markers of T cell activation and antigen presentation. We found a strong positive correlation between the expression of CD69 and HLA-DRA, markers of T cell activation, with the expression of all the genes except for OPTN and ETS1 in all breast cancer subtypes (Figure 4a). Similar findings were observed when markers of antigen presentation, namely CD40, CD86, and CD83, were evaluated (Figure 4b). Appendix A shows the data obtained using the cohort from the TCGA project.

### 2.5. CD47-Immune Signature Correlated with Gene Signatures of T Cell Activation

The findings described before suggested that the tumors expressing CD47 were enriched with T cells, DCs, and neutrophils. To further assess immune activation, we explored the correlation of our six-gene signature with already described genomic profiles of immune activation, including the HLA-A/B signature, the IFN gamma signature, the expanded immune gene signature, and the CTL signature [10,11,12]. We found that the CD47-immune signature was positively associated with these four transcriptomic profiles (Figure 4c), suggesting that this signature identifies tumors with a high presence of activated T cells and DCs. Given the fact that PTPRC or CD45 is a gene that codes for a marker globally expressed in different immune populations, being considered as a pan-leukocyte antigen, we repeated the analysis excluding this gene. As can be seen in Appendix Aa, the correlation was present even with the absence of this gene. 

### 2.6. Gene-Set Enrichment Analysis (GSEA) Confirm the Association of the CD47-Immune Signature with Pro-Tumoral Macrophages

We next used gene-set enrichment analysis (GSEA) to test whether the CD47-immune signature composed by PTPRC, HLA-E, TGFBR2, PTGER4, ETS1, and OPTN was preferentially associated with a specific macrophage polarization state. The CD47-immune signature was not significantly enriched in the transcriptome of either anti-inflammatory macrophages (M-MØ) or monocyte-derived pro-inflammatory (GM-MØ), suggesting that the expression of this gene set was independent of the macrophage polarization state (Figure 5a). However, the signature was found to be significantly enriched (FDR q value = 0.018) in the transcriptome of IL-10-treated adherent peripheral blood mononuclear cells (monocytes) (Figure 5b). Since IL-10 is a major factor that determines the pro-tumoral action of tumor-associated macrophages (TAM) [20,22], this result suggests that the 6-gene CD47-immune signature might be regulated by factors promoting TAMs. In a similar manner, we performed the same analysis but excluded PTPRC, observing that the results were in the same direction (Appendix A).

Last, we hypothesized that the genes that positively correlate with CD47 and that were associated with a good prognosis (see Figure 1 and Appendix A) might be associated with a specific type of macrophage polarization. To test this hypothesis, we analyzed the expression of this gene set on the ranked comparison of M-MØ and GM-MØ transcriptomes [21,23]. GSEA revealed a very significant enrichment of this set of genes in the transcriptome of pro-tumoral M-MØ (ES = 0.51; NES = 1.91; *p*-value = 0.000; FDR q-value = 0.000) (Figure 5c). Indeed, and using the available information on the transcriptomes of activated M-MØ and GM-MØ [24], a lower but significant enrichment was also observed in activated pro-tumoral M-MØ (ES = 0,33; NES = 1.31; *p*-value = 0.074; FDR q-value = 0.129) (Figure 5d).

Altogether, these results indicate that the expression of genes that positively correlate with CD47 and are associated with a good prognosis are preferentially expressed by macrophages with anti-inflammatory capacity.

## 3. Discussion

In the present article, we described a transcriptomic immune signature formed by six genes that were expressed in breast tumors with high expression of CD47 and were associated with favorable outcomes. The CD47 ligand is present on the surface of tumoral cells, and by binding to its receptor SIRPα, inhibits the induction of phagocytosis by macrophages [18,19,20]. CD47 is a perfect target to stimulate the activation of macrophages as well as other innate immunity cells, and several therapeutic strategies blocking the CD47- SIRPα axis are under evaluation in clinical studies. 

When analyzing the genes positively correlated to CD47 expression in breast cancer, we observed that only a very limited number of these genes, 33 (22.5%) predicted detrimental outcomes, while 83 genes (58.5%) predicted good outcomes. Among the identified functions of the genes linked to favorable prognosis, our attention was attracted to the two functions related to the immune system; the genes included within these functions were PTGER4, ETS1, PTPRC, HLA-E, TGFBR2, and OPTN. We termed the combination of these genes as the “CD47-immune signature”, and predicted favorable prognosis in all breast tumors, but particularly in the HER2+ and basal-like subtype. These results obtained from the exploratory dataset including 1764 patients was confirmed by an independent cohort that included 1947 patients. 

The CD47-immune signature includes a variety of genes that code for proteins with a wide range of functions that have been rarely described in relation to the immune response in cancer. PTGER4 which is a member of the G-protein coupled receptor family and can activate T cell factor signaling [25]; ETS1 which codes for a member of the ETS family of transcription factors that are involved in stem cell development, cell senescence and death, and tumorigenesis [26]; PTPRC a member of the protein tyrosine phosphatase (PTP) family that regulates a variety of cellular processes including cell growth, differentiation, mitosis, and oncogenic transformation [27]; HLA-E which belongs to the HLA class I heavy chain paralogues and functions as a ligand for natural killer (NK) cell inhibitory receptor KLRD1-KLRC1, enabling NK cells to monitor the expression of other MHC class I molecules in healthy cells [28]; TGFBR2 a transmembrane protein that has a protein kinase domain, forms a heterodimeric complex with TGF-beta receptor type-1, and binds TGF-beta [29]; and finally, OPTN that encodes the coiled-coil containing protein optineurin that interacts with adenovirus E3-14.7K protein and may utilize tumor necrosis factor-alpha or Fas-ligand pathways to mediate apoptosis, inflammation or vasoconstriction [30].

To study whether CD47 positive tumors are linked to an active T cell response, we correlated the CD47-immune signature with immune populations using bioinformatic approaches. Data demonstrated that there was a positive correlation at a single gene level with the presence of infiltrating T cells, DCs, and neutrophils. It is relevant to mention that although the strongest effect was observed in the basal-like and HER2+ population, such associations were also observed for all breast cancer patients. A limitation of our analysis was that we were not able to dissect the presence of immune infiltrates in the two different luminal breast cancer subtypes, luminal A and B. 

Finally, we aimed to confirm this association by exploring the correlation of each gene contained in the CD47-immune signature with well-known markers of T cell activation and of antigen presentation. We observed a positive correlation for all genes except for OPTN and ETS1. Moreover, a strong positive correlation was identified between the whole expression of the signature and that of described signatures of T cell activation, including HLA-A/B, the CTL signature, the expanded immune gene signature, and the IFN gamma signature [8,9,10,31]. Finally, a surprising finding was that no increase in the macrophage population was observed, but those identified were pro-tumoral. This finding was further confirmed when exploring the whole transcriptomic signature of each of the different subtypes of macrophages, in line with the inhibitory action of CD47. 

Our study had several limitations. This was an in silico study which needs further clinical validation using human samples. We do acknowledge that our study is based on associations and correlations among biological parameters. However, we tried to use different datasets as well as associations between genes, biomarkers, and immune populations to avoid findings produced by casualty. 

We believe it is relevant to explore if this signature could help identify patients that would respond to anti-CD47 agents. In line with this, the data observed here suggest the combined administration of anti-PD-L1 inhibitors with anti-CD47/ SIRPα agents to boost the T cell response and the activation of macrophages. In this regard, ongoing studies are exploring the activity of targeting both pathways [16,17]. Given the fact that strategies are under evaluation targeting CD47 and SIRPα, it might be interesting to evaluate the transcriptomic profile of tumors with high expression of SIRPα.

## 4. Materials and Methods

### 4.1. Exploratory Cohort

Samples included in the KM Plotter Online Tool (http://www.kmplot.com, accessed on 20 March 2020) [32] were used as an exploratory cohort. This publicly available database shows the relationship between gene expression and patient outcome in different breast cancer subtypes, including relapse-free survival (RFS) and overall survival (OS). Patients were distributed according to the best cutoff values of the gene expression (lowest *p*-value) into “high” vs. “low”. RFS was defined as the time from diagnosis to the first recurrence, and OS as the time from diagnosis to patient death. The number of breast cancer patients included in each subtype for RFS was: all: *n* = 1764; basal-like: *n* = 360; HER2+: *n* = 156; luminal B: *n* = 407; luminal A: *n* = 841. For OS: all: *n* = 626; basal-like: *n* = 156; HER2+: *n* = 73; luminal B: *n* = 129; luminal A: *n* = 271 (HGU133 array 2.0) (available data April 2020).

To identify genes whose expression correlated to CD47 expression, the probe set 226016 was correlated using all samples from the exploratory cohort. For each gene, Spearman rank correlation was computed to compare its normalized gene expression and CD47 expression. Then, the genes were ranked based on the achieved Spearman correlation coefficients.

### 4.2. Validation Cohort

Survival analysis was performed in basal-like (*n* = 331), HER2+ (*n* = 135), luminal B (*n* = 668), and luminal A (*n* = 825) patient samples from the METABRIC (Molecular Taxonomy of Breast Cancer International Consortium) project (PMID: 22522925). Gene expression values were quantile normalized in R. The molecular subtype designation was based on St. Gallen criteria [33], and the expression of ESR1, ERBB2, and MKI67 on the arrays was used to define the patient cohorts. In this, basal breast cancer was defined by those negative for ESR1 and ERBB2, luminal A was defined as those ESR1 positive ERBB2 negative MKI67 negative, HER2+enriched was defined as those ERBB2 positive ESR1 negative, and luminal B comprises all remaining samples.

### 4.3. Gene Function Analysis

Genes positive-correlated with CD47 and associated with good or detrimental outcomes were analyzed using the biological function enrichment analyses tool Enrichr (http://www.amp.pharm.mssm.edu/Enrichr/, accessed on 20 March 2020) [34]. Biological process with a *p*-value < 0.01 were selected for CD47 positively correlated genes and with a *p*-value < 0.05 CD47 negatively correlated genes. Biological processes related to the immune system were grouped. Immune system processes contained the following gene ontologies: positive regulation of defense response (GO:0031349), regulation of T cell-mediated cytotoxicity (GO:0001914), and positive regulation of alpha-beta T cell proliferation (GO:0046641).

### 4.4. Protein Functional Analyses

We used UniProt Online Tool (http://www.uniprot.org/, accessed on 20 March 2020) [35] for protein functional analysis. UniProt provides a comprehensive, high-quality, and freely accessible resource of protein sequence and functional information. All data is freely available on the web. Proteins codified by genes related to immune system processes were studied. For complete protein functional analyses, we collected all biological function gene ontologies included in UniProt.

### 4.5. Association between Tumor Immune Infiltrates and Gene Expression

Tumor Immune Estimation Resource (TIMER) platform (https://cistrome.shinyapps.io/timer/, accessed on 2 April 2020) [36] was used to analyze tumor purity, and the association between the presence of tumor immune infiltrates, namely CD4+ T cells, CD8+ T cells, DCs, macrophages, neutrophils, B cell, and macrophages and the expression of the selected genes. TIMER contains 10,897 samples from diverse cancer types from the TCGA (The Cancer Genome Atlas) project. We explored the tumor immune infiltrates in breast cancer subtypes: basal-like, HER2+, and luminal. TIMER does not allow for the analysis of the luminal A and luminal B subtypes separately. 

### 4.6. Correlation between Gene Expression and T Cell Activation and Antigen Presentation

CANCERTOOL (http://web.bioinformatics.cicbiogune.es/CANCERTOOL/index.html, accessed on 2 April 2020) [37] was used to explore the relationship between the expression of CD47-immune signature genes and the expression of T cell activation (CD69 and HLA-DRA) and antigen presentation (CD40, CD86, and CD83) markers in all, basal-like, HER2+, luminal B, and luminal A breast cancer. This open-access resource for the analysis of gene expression provides the Pearson correlation coefficients of every pair of genes analyzed. The datasets used for the analysis included METABRIC as the primary cohort, and TCGA, as a validation cohort.

The complete METABRIC cohort (total: *n* = 1988; basal: *n* = 331; HER2+: *n* = 135; luminal B: *n* = 668; luminal A: *n* = 825 breast cancer) was used to explore the correlation between the identified CD47-immune signature and previously described signatures, including: the HLA signature (HLA-A and HLA-B) [10] the interferon (IFN) gamma signature (IDO1, CXCL10, CXCL9, HLA-DRA, ISGF-3, and IFNG) [12], the expanded immune gene signature (CD30, IDO1, CIITA, CD3E, CCL5, GZMK, CD2, HLA-DRA, CXCL13, IL3RG, NKG7, HLA-E, CXCR6, LAG3, TAGAP, CXCL10, STAT1, and GZMB) [29] and the cytotoxic T lymphocyte (CTL) signature (CD8A, CD8B, GZMA, GZMB, and PRF1) [11]. 

### 4.7. Correlation between Gene Expression and Macrophage Signatures

Gene set enrichment analysis (GSEA) (http://software.broadinstitute.org/gsea/index.jsp, accessed on 5 May 2020) [38] was done to assess enrichment of the indicated gene-sets in the transcriptomes of human monocyte-derived pro-inflammatory (GM-MØ) and anti-inflammatory macrophages (M-MØ) (GSE27792 and GSE68061) [20,22].

### 4.8. Statistical Analysis

The Kaplan–Meier (KM) plots are presented with the hazard ratio (HR), the 95% confidence interval (CI), the log-rank *p*-value (*p*), and the false discovery rate (FDR). The FDR was computed after performing the Cox regression analysis across all cutoff values between the lower and upper quartiles of expression, and only results with an FDR below 10% were accepted as significant. For the METABRIC dataset, only the median was used to define high and low cohorts, and therefore FDR values were not calculated. In addition to the HR and FDR cutoff values described above, statistical significance was defined as *p* < 0.05. Genes that had an HR < 1 and a *p* < 0.05 were considered predictors of a favorable outcome, while genes that had an HR > 1 and *p* < 0.05 were considered predictors of detrimental outcome.

## 5. Conclusions

Here we described an immune gene signature associated with elevated levels of CD47 that predicts favorable outcomes in breast cancer tumors. In addition, the described signature was linked with the presence of T cell, DC, and neutrophil infiltrates, T cell activation and antigen presentation, and correlated with pro-tumoral macrophages. Further studies should confirm the predictive capacity of this signature in ongoing clinical studies. 

## Figures and Tables

**Figure 1 ijms-22-03836-f001:**
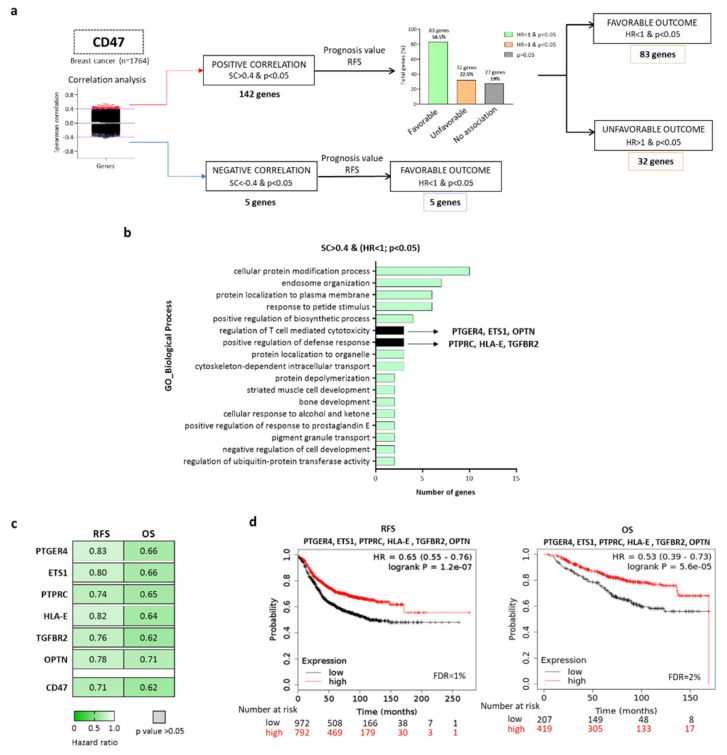
Immune system related transcriptional profiles associated with outcome and with CD47 expression in breast cancer. (**a**) Flow chart of gene selection, describing the tools and selection criteria used. (**b**) Functional analyses of the selected genes described in A using Enrichr Online Tool. For positively correlated genes (Spearman correlation (SC) > 0.4 and Hazard Ratio (HR) < 1), gene ontologies (GO) of biological process with a *p* < 0.05 are shown. The processes related to the immune system are highlighted. (**c**) Heat map displaying HR values extracted from Kaplan–Meier survival plots of the association between PTGER4, ETS1, PTPRC, HLA-E, TGFBR2, OPTN, and CD47 individually expressed and patient prognosis, including relapse-free survival (RFS) (*n* = 1764) and overall survival (OS) (*n* = 626), for all breast subtypes from the exploratory cohort. (**d**) Kaplan–Meier survival plots of the association between PTGER4, ETS1, PTPRC, HLA-E, TGFBR2, and OPTN mean expression levels and patient prognosis, including relapse-free survival (RFS) (*n* = 1764) and overall survival (OS) (*n* = 626) for all breast subtypes from the exploratory cohort.

**Figure 2 ijms-22-03836-f002:**
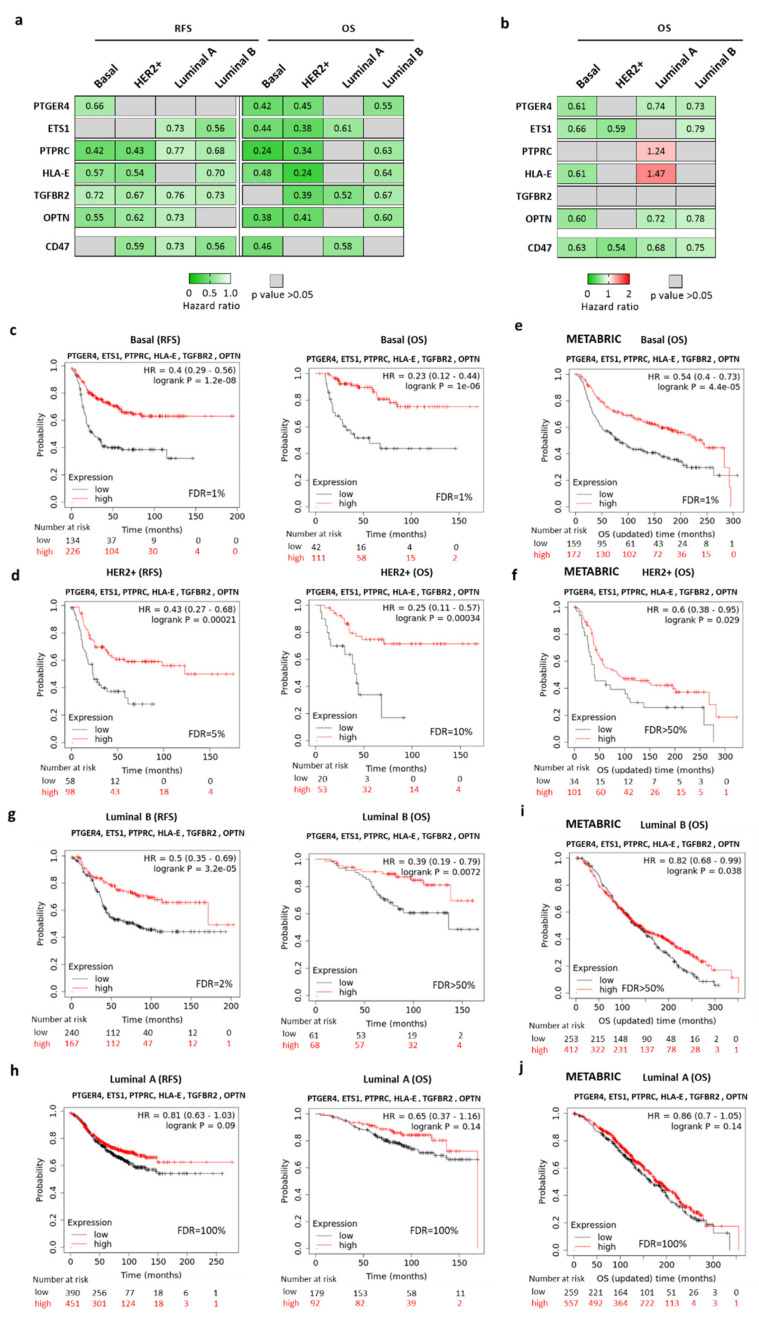
Kaplan–Meier survival curves of the association between transcriptomic expression of PTGER4, ETS1, PTPRC, HLA-E, TGFBR2, and OPTN and clinical outcome in basal-like, HER2+, luminal A, and luminal B breast cancer patients. (**a**,**b**) Heat map displaying HR values extracted from Kaplan–Meier survival plots for the association between PTGER4, ETS1, PTPRC, HLA-E, TGFBR2, OPTN, and CD47 individually expression levels and patient prognosis, in the exploratory cohort and Molecular Taxonomy of Breast Cancer International Consortium (METABRIC) validation cohort, respectively. (**c**,**d**) basal-like (relapse-free survival (RFS); *n* = 360 and OS; *n* = 153), and HER2+ (RFS; *n* = 156 and OS; *n* = 53), breast tumors in the exploratory cohort. (**e**,**f**) basal-like (OS; *n* = 331) and HER2+ (OS; *n* = 135) in the validation cohort (METABIRC project). (**g**,**h**) luminal B (RFS; *n* = 841 and OS; *n* = 271), and luminal A (RFS; *n* = 407 and OS; *n* = 129) in the exploratory cohort. (**i**,**j**) luminal B (OS; *n* = 816), and luminal A (OS; *n* = 665) breast tumors in the validation cohort (METABIRC project).

**Figure 3 ijms-22-03836-f003:**
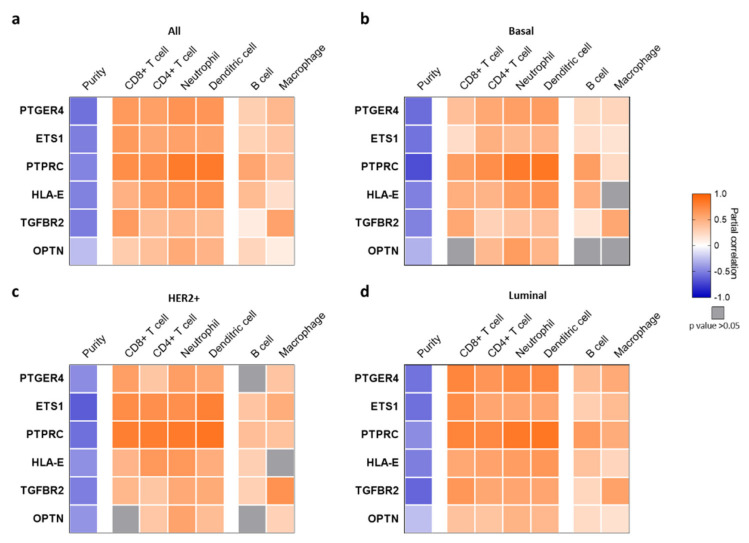
Association of the expression of the selected genes with immune infiltrates in breast cancer. Heat map depicting the Pearson correlation coefficient (R) between gene expression (PTGER4, ETS1, PTPRC, HLA-E, TGFBR2, and OPTN), tumor purity, and the presence of tumor immune infiltrates in (**a**) all, (**b**) basal-like, (**c**) HER2+, and (**d**) luminal breast cancer tumors using TIMER. Tumor immune infiltrates were separated into two groups: first, CD8+ T cells, CD4+ T cells, neutrophils, and dendritic cells (DCs); second, B cells and macrophages.

**Figure 4 ijms-22-03836-f004:**
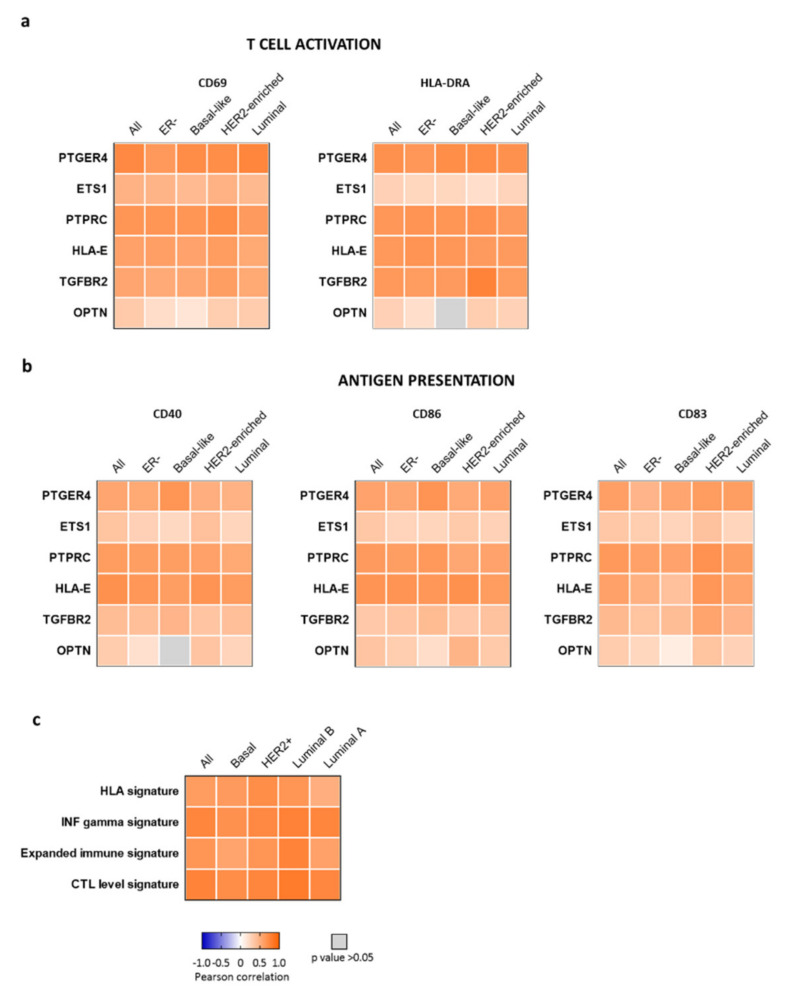
Relationship between gene expression and genomic signatures of immune activation. Heat map depicting the Pearson correlation coefficient (R) of the association between (**a**) markers of T cell activation (CD69 and HLA-DRA) or (**b**) antigen presentation (CD40, CD86, and CD83) and the expression of the selected genes using CANCERTOOL and the METABRIC cohort. (**c**) Heat map of Pearson correlation coefficient (R) of the expression of the CD47-immune signature and the HLA signature, IFN gamma signature, expanded immune gene signature, and CTL level signature in all (*n* = 1988), basal (*n* = 334), HER2+ (*n* = 137), luminal B (*n* = 680), and luminal A (*n* = 837) breast cancer.

**Figure 5 ijms-22-03836-f005:**
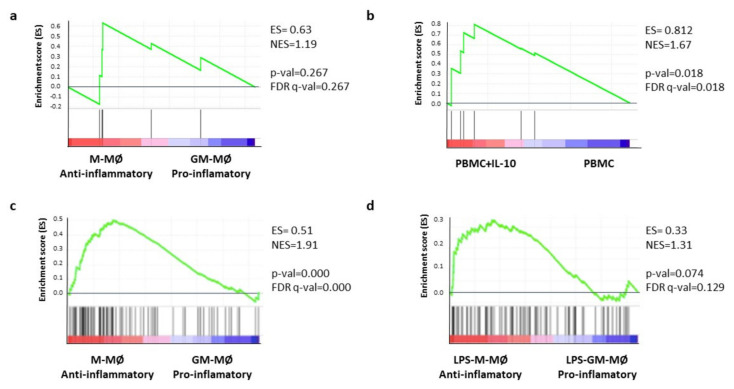
Gene-set enrichment analysis (GSEA) between gene expression and macrophage signatures. Gene-set enrichment analysis (GSEA) of the six-gene CD47-immune signature on (**a**) the ranked comparison of anti-inflammatory macrophages (M-MØ) or M2 and monocyte-derived pro-inflammatory (GM-MØ) or M1 whole transcriptomes, previously described in [20,22] (GSE27792 and GSE68061) or (**b**) the transcriptomes of adherent human peripheral blood mononuclear cells either untreated (PBMC) or treated with 10 ng/mL IL-10 for 24 h (PBMC + IL-10) that have been previously described. GSEA of the genes that positively correlate with CD47 and are associated with good prognosis (Appendix A) on (**c**) the ranked comparison of M-MØ or M2 and GM-MØ or M1 whole transcriptomes, previously described in [21,23] (GSE27792 and GSE68061) or (**d**) the ranked comparison of the transcriptome of lipopolysaccharide (LPS)-treated M-MØ or M2 and LPS-treated GM-MØ or M1 transcriptomes, previously described in [24] (GSE99056).

## Data Availability

Data are available upon reasonable request to the corresponding author. Part of the data that support the findings of this study are publicly available in the Gene Expression Omnibus (GEO) repository; data set accession numbers: GSE27792, GSE68061, GSE99056, and GSE84622.

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
