# Peer review of "Transcriptomic Profiles of CD47 in Breast Tumors Predict Outcome and Are Associated with Immune Activation"

_ijms, 2021, doi:10.3390/ijms22083836_

Round 1

Reviewer 1 Report

Targeting CD47 immunotherapy demonstrates an anti-tumor efficacy in human cancers. Based on current data set, authors explored the role of CD47 in breast cancer and analyzed correlation between  surface markers expressed on Tumor infiltrated macrophages and CD47 expression. Related concerns must be addressed as following;

  1. Novelty  The role of CD47 in human cancer immunotherapy has been widely explored by other research groups, related conclusions can be obtained in previous studies, thus the current  manuscript lacks novelty, in other words, just reaffirmation or supplement of previous research.
  2. Validation of dataset analysis

      Authors performed related analysis using current dataset, however, regarding to related conclusions, validation experiments are necessary, "wet" bench work could validate corresponding results.

Author Response

Targeting CD47 immunotherapy demonstrates an anti-tumor efficacy in human cancers. Based on current data set, authors explored the role of CD47 in breast cancer and analyzed correlation between  surface markers expressed on Tumor infiltrated macrophages and CD47 expression. Related concerns must be addressed as following;

  1. Novelty The role of CD47 in human cancer immunotherapy has been widely explored by other research groups, related conclusions can be obtained in previous studies, thus the current  manuscript lacks novelty, in other words, just reaffirmation or supplement of previous research.
  2. Validation of dataset analysis

      Authors performed related analysis using current dataset, however, regarding to related conclusions, validation experiments are necessary, "wet" bench work could validate corresponding results.

Response:

We still consider that the article is novel and could be of interest for the readers of the journal. Identification of genomic correlates of response to immunotherapies is a great area in cancer research, and in the case of agents targeting CD47, no biomarker of response exists. In addition, although some immunologic basic science information described here is known, the relation of that information with the presence of CD47 has not been described before, and therefore, merits to be shown to the scientific community.  

Regarding the validation dataset it is worth mentioning that we have included two different datasets, an exploratory and a validation, both using a large number of patients from different cohorts of patients.  

Briefly, the KM Plotter Online Tool (http://www.kmplot.com) were used as exploratory cohort (number of patient for RFS was: all n=1764, Basal-like n=360, HER2+ n=156, Luminal B n=407, and Luminal A n=841, and number of patient for OS was: all n=626, Basal-like n=156, HER2+ n=73, Luminal B n=129, and Luminal A n=271) (Figure 1c,d and Figure 2a,c,d,g,h). In addition the patient samples from the METABRIC (Molecular Taxonomy of Breast Cancer International Consortium) project (PMID: 22522925) were used as validation cohort (number of patient; Basal-like (n=331), HER2+ (n=135), Luminal B (n=668), and Luminal A (n=825) (Figure 2b,e,f,i,j).

The evaluation of our results in patients would need to get tumor data from patients with a minimum follow up for analysis. We acknowledge that this will strength the results but this process will take a very long period of time. To this regard we are currently working on exploring this data using cancer patient samples.

Reviewer 2 Report

In their study, Noblejas-López et al. explore genomic databases for expression of CD47, and describe genes that are either positively or negatively associated with CD47 mRNA. They then attempt to generate a predictive gene signature based on this information, and relate it to clinical outcomes and mRNA expression of genes involved in immune cell processes.

Though ambitious and potentially interesting, this study is almost entirely descriptive, merely showing correlations of unknown significance without any true evidence of causation. Additionally, there are several issues with their study design that regrettably limit enthusiasm at this time.

Most notably, the authors also make several misleading claims regarding the role of the selected genes in their panel, and many assumptions regarding immune activation that are cause for concern. Additionally, the data is often presented in a way that is convoluted and difficult to interpret. Finally, the relationship between their panel and genes involved in immune activation is of grave concern, as their panel contains a pan-leukocyte marker, likely a massive confounding factor. Specific comments on how to improve their work are listed below

Figure 1A) It is unclear why the authors are focusing on the gene sets that only show 3 correlated genes, while others show up to 10. This appears to be a largely biased method that is not adequately explained in the text.

Figure 1C) the stated “principal functions” of these genes are extremely misleading when listed out of context, and because of this, the reason for selecting the 6 genes that they did again seems rather arbitrary. Though the authors attempted to offer additional insight into the potential role for these genes in Table S1, Figure 1C this is still of major concern and needs to be redone or removed altogether. As an example, TGFBR2 is listed as being a driver of alpha-beta cell proliferation. However, TGFB signals are an extremely well-established mediator of immune escape in cancer, functioning to promote T-cell anergy, induce peripheral T-regs, etc. This is without question its dominant role in breast cancer immunology, particularly in light of evidence supporting combined TGFBR and PD-1 inhibition in breast and other cancers.  

Figure 1D) The importance of these plots and all subsequent survival data based on the expression of these genes are limited by the issues described above. Additionally, the authors should show the predictive value of each individual genes, and CD47.

Table 1) This table is extremely cumbersome and I find myself constantly revisiting the legend to make sense of it. This content must be shown more clearly and better labeled. Also, as it is not central to the overarching conclusions of the manuscript, it may also be better if moved to the supplement.

Figure 2) As previously, the importance of these KM plots are limited by the issues described in Figure 1, namely the seemingly arbitrary selection of genes. Additionally, the authors do not show the predictive value of either CD47 or each gene individually.

Figure 3-5) Here, the authors attempt to relate their gene signature to T-cell activation. This is a fundamentally flawed design, as their panel includes the gene PTPRC, also known as CD45. This is a pan-leukocyte antigen that will absolutely correlate with markers for all other immune cell types and surface markers. Hence, it is extremely unlikely that this panel predicts for immune activation, but rather that it simply includes a leukocyte surrogate marker. I all but guarantee that the authors will find an association between any gene panel that includes PTPRC and markers of T-cell activation, T-cell exhaustion, M1 polarization, M2 polarization, or any other immune process they choose. 

Author Response

In their study, Noblejas-López et al. explore genomic databases for expression of CD47, and describe genes that are either positively or negatively associated with CD47 mRNA. They then attempt to generate a predictive gene signature based on this information, and relate it to clinical outcomes and mRNA expression of genes involved in immune cell processes.

Though ambitious and potentially interesting, this study is almost entirely descriptive, merely showing correlations of unknown significance without any true evidence of causation. Additionally, there are several issues with their study design that regrettably limit enthusiasm at this time.

Most notably, the authors also make several misleading claims regarding the role of the selected genes in their panel, and many assumptions regarding immune activation that are cause for concern. Additionally, the data is often presented in a way that is convoluted and difficult to interpret. Finally, the relationship between their panel and genes involved in immune activation is of grave concern, as their panel contains a pan-leukocyte marker, likely a massive confounding factor. Specific comments on how to improve their work are listed below:

Figure 1A) It is unclear why the authors are focusing on the gene sets that only show 3 correlated genes, while others show up to 10. This appears to be a largely biased method that is not adequately explained in the text.

Response:

We apologies for the limited explanation provided in the text. We have selected these two functions as were the only ones related to the immune system, as explained in the materials and methods section (lines 377-380). Although we acknowledge that the number of genes included in these functions are smaller than in other groups, we aimed to select only those associated with the immune system. To improve the clarity, we have explained with more detailed this approach in the results section (highlighted, lines 123-125).

Figure 1C) the stated “principal functions” of these genes are extremely misleading when listed out of context, and because of this, the reason for selecting the 6 genes that they did again seems rather arbitrary. Though the authors attempted to offer additional insight into the potential role for these genes in Table S1, Figure 1C this is still of major concern and needs to be redone or removed altogether. As an example, TGFBR2 is listed as being a driver of alpha-beta cell proliferation. However, TGFB signals are an extremely well-established mediator of immune escape in cancer, functioning to promote T-cell anergy, induce peripheral T-regs, etc. This is without question its dominant role in breast cancer immunology, particularly in light of evidence supporting combined TGFBR and PD-1 inhibition in breast and other cancers. 

Response:

We consider that although we have only selected those six genes, these set of genes were obtained for the gene ontology analysis and no cherry picking selection was performed. 83 genes were selected based on a strong correlation score and a clear association with favorable survival. From all those 83, only 6 were included in biological functions linked with immunotherapy properties. So we consider that although the approach have limitations, the methods and the results are valid, and no intentional bias exist. However, to improve the clarity of the manuscript we have deleted the figure 1C. Supplementary table 1 includes the functions of proteins coded by these genes.

Figure 1D) The importance of these plots and all subsequent survival data based on the expression of these genes are limited by the issues described above. Additionally, the authors should show the predictive value of each individual genes, and CD47.

Response:

We agree about this suggestion and we have included this information in the current figure 1C. New sentences in the results section of the manuscript has been added (highlighted, lines 136-138 and lines 146-147). We want to remark that CD47-immune signature prediction was better than single gene prediction including CD47, and displayed a very low FDR.

Table 1) This table is extremely cumbersome and I find myself constantly revisiting the legend to make sense of it. This content must be shown more clearly and better labeled. Also, as it is not central to the overarching conclusions of the manuscript, it may also be better if moved to the supplement.

Response:

We agree with this comment and we appreciate the suggestion. We have moved this table to the supplementary information as new supplementary figure 1.

Figure 2) As previously, the importance of these KM plots are limited by the issues described in Figure 1, namely the seemingly arbitrary selection of genes. Additionally, the authors do not show the predictive value of either CD47 or each gene individually.

Response:

We agree with this comment. We are now showing the predictive value of each gene as figure 2a and figure 2b. A sentence in the results section of the manuscript has been added (highlighted, lines 149-154 and lines 166-167). We want to remark that CD47-immune signature prediction was better than single gene prediction including CD47 and displayed a very low FDR.

Figure 3-5) Here, the authors attempt to relate their gene signature to T-cell activation. This is a fundamentally flawed design, as their panel includes the gene PTPRC, also known as CD45. This is a pan-leukocyte antigen that will absolutely correlate with markers for all other immune cell types and surface markers. Hence, it is extremely unlikely that this panel predicts for immune activation, but rather that it simply includes a leukocyte surrogate marker. I all but guarantee that the authors will find an association between any gene panel that includes PTPRC and markers of T-cell activation, T-cell exhaustion, M1 polarization, M2 polarization, or any other immune process they choose.

Response:

To demonstrate that the signature is still valid independently of the presence of PTPRC we have delete this gene from the combined analysis.

The analysis included in figure 3 and figure 4 a and b, is independent for each gene. Here we can see the correlation for each gene in the all breast cancer group, and by breast cancer subtype. The correlation of each gene with immune infiltrates is displayed in figure 3. In a similar manner the correlation of each gene with other markers of T cell activation is displayed in figure 4a and with antigen presentation markers in figure 4b.

In figure 4c we have repeated the analysis excluding the PTPRC gene as suggested by the reviewer and we have included this data as supplementary figure 4a. It can be observed that the correlation of the signature is still present validation the results. A sentence in the results section of the manuscript has been added (highlighted, lines 236-239).

In a similar manner as in figure 5a and 5b, we have repeated the analysis excluding the PTPRC gene. As can be seen in supplementary figure 4b and 4c results are in the same direction, without varying the findings. A sentence in the results section of the manuscript has been added (highlighted, lines 252-254).

The results displayed here demonstrate that the weight of the signature is given by the combination of the six genes and the exclusion of PTPRC does not modify the findings.

Round 2

Reviewer 1 Report

Considering the academic value of this manuscript, authors would better add wet-bench research data.

Author Response

We agree with this consideration. However, to execute this set of experiments would take a very long time, and it is impossible at this moment to perform and include all of them in this article. However, we are working on this for a next work.

Reviewer 2 Report

The authors have done an admirable job incorporating the extensive suggestions from the previous round of review. 

Author Response

We appreciate this comment.